# Exploring Opportunities to Enhance the Screening and Surveillance of Hepatocellular Carcinoma in Non-Alcoholic Fatty Liver Disease (NAFLD) through Risk Stratification Algorithms Incorporating Ultrasound Elastography

**DOI:** 10.3390/cancers15164097

**Published:** 2023-08-14

**Authors:** Madalina-Gabriela Taru, Monica Lupsor-Platon

**Affiliations:** 1Hepatology Department, Regional Institute of Gastroenterology and Hepatology “Octavian Fodor”, 400162 Cluj-Napoca, Romania; madalinagabriel.tar2@unibo.it; 2“Iuliu Hatieganu” University of Medicine and Pharmacy, 400012 Cluj-Napoca, Romania; 3Medical Imaging Department, Regional Institute of Gastroenterology and Hepatology “Octavian Fodor”, 400162 Cluj-Napoca, Romania

**Keywords:** hepatocellular carcinoma, NAFLD, MASLD, risk stratification, vibration-controlled transient elastography, ultrasound elastography

## Abstract

**Simple Summary:**

Non-alcoholic fatty liver disease (NAFLD) has become a significant public health concern, affecting over 30% of the population. The incidence of NAFLD-related hepatocellular carcinoma (HCC) has been rising, with unique pathogenic factors compared to other causes of HCC. Early detection is crucial for better outcomes. Currently, it is still unclear which subset of non-cirrhotic NAFLD-patients could benefit from HCC screening, and a better characterization and stratification of this population is required. This review underscores the potential of liver ultrasound elastography as a risk assessment tool for HCC development in NAFLD. Alongside exploring the potential advancement of non-invasive tools and algorithms for effectively stratifying HCC risk in NAFLD, we offer essential insights that could enable readers to enhance the personalized assessment of NAFLD-related HCC risk in a more systematic manner.

**Abstract:**

Non-alcoholic fatty liver disease (NAFLD), with its progressive form, non-alcoholic steatohepatitis (NASH), has emerged as a significant public health concern, affecting over 30% of the global population. Hepatocellular carcinoma (HCC), a complication associated with both cirrhotic and non-cirrhotic NAFLD, has shown a significant increase in incidence. A substantial proportion of NAFLD-related HCC occurs in non-cirrhotic livers, highlighting the need for improved risk stratification and surveillance strategies. This comprehensive review explores the potential role of liver ultrasound elastography as a risk assessment tool for HCC development in NAFLD and highlights the importance of effective screening tools for early, cost-effective detection and improved management of NAFLD-related HCC. The integration of non-invasive tools and algorithms into risk stratification strategies could have the capacity to enhance NAFLD-related HCC screening and surveillance effectiveness. Alongside exploring the potential advancement of non-invasive tools and algorithms for effectively stratifying HCC risk in NAFLD, we offer essential perspectives that could enable readers to improve the personalized assessment of NAFLD-related HCC risk through a more methodical screening approach.

## 1. Introduction

Non-alcoholic fatty liver disease (NAFLD), and its progressive form, non-alcoholic steatohepatitis (NASH), has become a major public health problem, affecting more than 30% of the population worldwide [1,2]. Hepatocellular carcinoma (HCC), a complication associated with NAFLD, is characterized by an incidence that has risen significantly over recent decades, making it one of the most common cancers worldwide [3].

The presence of liver cirrhosis increases the risk of HCC in patients with NAFLD, as it does in other types of liver disease [4]. Nevertheless, patients with NAFLD but without cirrhosis, yet susceptible to developing HCC, tend to present distinct features such as diabetes or insulin resistance, obesity, and older age [5,6,7,8].

A particular yet critical clinical feature of NAFLD- and NASH-related HCC is the greater tendency for tumors to emerge before cirrhosis is established [9,10,11]. Regarding the presence and the evolution of liver fibrosis, it has been recently outlined that HCC incidence increases along with the progression of NAFLD, from simple steatosis (0.8 per 1000 person-years) to non-cirrhotic fibrosis (2.3 per 1000 person-years), and NAFLD-related cirrhosis (6.2 per 1000 person-years), respectively [12]. The same progressive trend of increased HCC incidence, together with the progression of the liver disease, has been reported in another study; increased incidence from those with simple steatosis (0.44 per 1000 person-years) to those with NASH (5.29 per 1000 person-years) [13].

On the other hand, NAFLD presents an incidence that is increasing in parallel to the rise of obesity, diabetes, and metabolic syndrome [1,14,15]. The presence of these comorbid conditions contributes to an altered lipid metabolism and chronic inflammation, favoring a pro-carcinogenic environment and setting the ground for liver cancer development [16,17].

In this context, NAFLD has become the leading cause of HCC in the absence of cirrhosis, with approximately 30% of NAFLD-related HCC occurring in non-cirrhotic livers, compared with around 10% of non-cirrhotic HCC in other liver diseases [9,10,18].

The NAFLD-related HCC trends are definitely concerning: NAFLD-related HCC has increased in both incidence and mortality [19]; NASH is the fastest growing cause of HCC in liver transplant (LT) candidates [20]; despite the older age of patients with NAFLD-HCC and the presence of comorbidities, HCC has become more common in patients transplanted for NAFLD compared with other etiologies [21]. Considering these trends, there is a rising concern that in the future, these figures will eventually result in a considerably higher burden of NAFLD-related HCC.

Among the patients with NAFLD, it can be estimated that 10–20% might develop hepatic inflammation and progress to non-alcoholic steatohepatitis (NASH) and that in the global population, approximately 3–5% have NASH [14]. In the context of NAFLD and NASH, within recent years, fibrosis has consistently emerged as the most important histologic feature predicting clinical events [22,23]. Even though the build-up of hepatic fibrosis takes at least 20 years, or even longer in the setting of NAFLD [24], as the prevalence of overweight and obesity are increasing in younger age groups [25,26], and considering the fact that modern treatments for cardiovascular diseases have improved life expectancy [27], there is also more time for the NAFLD population to develop NASH, advanced fibrosis (≥F3), cirrhosis (F4), and cirrhosis related complications, including HCC.

As a significant fraction of HCC develops in non-cirrhotic NAFLD, new clinical challenges arise as these patients are usually not enrolled in HCC surveillance programs [28].

Discovering the most suitable screening techniques, capable of directing risk assessment and frequency of monitoring for NAFLD-related HCC, is a central focus of interest among the scientific community [29,30]. We believe that by considering specific risk factors, along with a deeper comprehension of the underlying liver disease, evaluated through non-invasive methods, the detection of HCC linked to non-cirrhotic NAFLD could be improved. Nevertheless, NAFLD stands as a complex, dynamic, heterogeneous, and multifactorial disease, so personalized and precision medicine should also be considered when making management and treatment decisions.

In the following, we provide a comprehensive review addressing HCC risk in patients with NAFLD, and we underscore essential considerations for clinical practice. We bring into focus the potential role of liver ultrasound elastography as a risk assessment tool for the development of HCC in NAFLD and its likelihood of being integrated into multiparametric decision-making approaches. This review will concentrate on characterizing the liver parenchyma and identifying the possible risk factors linked to the occurrence of HCC within the framework of NAFLD. However, the assessment of the focal liver lesion will not be discussed in this review.

Alongside exploring the potential advancement of non-invasive tools and algorithms for effectively stratifying HCC risk in NAFLD, we offer essential insights that could enable readers to enhance the personalized assessment of NAFLD-related HCC risk in a more systematic manner.

## 2. Screening for NAFLD-Related HCC and Possibilities for Optimizing Strategies

Cancer surveillance refers to the use of biochemical or imaging modalities to detect early forms of malignancy. Cost-effectiveness analyses have indicated that an incidence of 1.5% per year or more would justify HCC surveillance in cirrhotic patients, irrespective of the underlying etiology [31,32]. Nevertheless, recent modeling studies have demonstrated that an annual HCC incidence of approximately 0.8% would be a cost-effective threshold for initiating surveillance in patients with cirrhosis [33].

At present, society guidelines recommend HCC primary surveillance for patients with chronic hepatitis B (CHB) and cirrhosis, based on abdominal ultrasound (US) evaluation with or without serum alpha-fetoprotein (AFP) levels [34,35,36]. Nevertheless, the role of AFP in HCC surveillance has been controversial. European guidelines do not recommend the use of AFP for surveillance because of its poor sensitivity for the detection of early-stage HCC, as long as the American guidelines still endorse the use of AFP as it may increase the detection performance of the US [37]. Unfortunately, current HCC surveillance strategies remain suboptimal [38], and in addition to this, patients with NAFLD-associated cirrhosis usually tend to be under-screened compared to other etiologies [39]. With a low implementation rate (<25%) and considering the suboptimal performance of ultrasound and AFP for early detection of HCC (63% sensitivity for early stages), there is an urgent need to develop new tools for HCC surveillance and to increase the adherence to the screening programs [37,40,41].

As early-stage diagnosis improves survival, the current clinical practice guidelines recommend screening for HCC in patients with cirrhosis due to NAFLD [35,36] but do not support the routine utilization of HCC surveillance in patients with non-cirrhotic NAFLD. At present, in the setting of NAFLD, it is stated that the best benefit of HCC screening could be attained among those patients with compensated cirrhosis or with decompensated cirrhosis listed for liver transplantation [29]. Nevertheless, surveillance might be considered in selected patients with advanced fibrosis (≥F3) on a case-by-case basis [42], particularly for those in whom there is clinical suspicion for under-staging of fibrosis by noninvasive markers or biopsy [43]. The current EASL practice guidelines suggest that for non-cirrhotic F3 patients, regardless of etiology, surveillance may be considered based on an individual risk assessment [36].

In recent years, the importance of US surveillance for NAFLD-related HCC has gained prominence, as even for patients with simple steatosis, high rates of NAFLD-associated HCC have been reported [28,44,45]. Despite evidence that NAFLD-associated HCC may arise in the absence of cirrhosis in a significant number of cases, its management is proving to be quite challenging at present. Amidst the current challenges, the following becomes apparent: NAFLD-related HCC is often diagnosed at advanced stages [46], it is associated with a decreased likelihood of receiving curative therapy, including liver transplantation [47], and it is associated with poorer survival [29]. Nevertheless, it is still unclear whether the risk of HCC in NAFLD patients without cirrhosis is high enough to render HCC surveillance cost-effective. According to some authors, NAFLD-related HCC occurs in the absence of cirrhosis; however, these patients have a very low annual HCC incidence of 0.008 per 100 person-years, so surveillance is not cost-effective in this population [48,49], but patients with NAFLD without cirrhosis, particularly those with advanced fibrosis (≥F3), could benefit from risk stratification tools to identify those at highest risk to whom surveillance could be targeted in the future [43].

Currently, several promising biomarkers are being evaluated for HCC surveillance, although most still require further validation [50,51,52].

In summary, within the current context, NAFLD patients represent a special population for which optimization of HCC screening modalities is definitely needed, and additional resources should be invested in order to address this issue adequately [38].

From our perspective, conducting NAFLD-related HCC screening in the future could be extended to well-identified individuals who are at risk. By accurately defining this particular population, the screening methods could also become more cost-effective. In order to provide a synthetic and practical approach to a possible portrayal of the at-risk population, we believe that its attributes could be condensed into two main categories, such as factors related to the severity of liver disease (specific, local, intrinsic factors) while the second category could involve “overall, systemic, extrinsic” factors, including the comorbid conditions associated with NAFLD. The blending of these factors could facilitate the future advancement of risk stratification strategies and algorithms that could provide guidance for NAFLD-related HCC surveillance in clinical practice.

## 3. Risk Factors Associated with NAFLD-Related Hepatocellular Carcinoma

In the setting of non-cirrhotic NAFLD, HCC surveillance could be performed in at-risk individuals [53,54] and might be targeted to those potentially eligible for curative treatment with the scope of improving survival [43].

As briefly outlined above, the onset of HCC in NAFLD patients is the result of multiple interactions between various factors such as the underlying liver condition, ethnicity, genetics, the presence of comorbidities, gut microbiome, environmental factors, and underlying factors such as the immune and inflammatory responses, oxidative stress, deoxyribonucleic acid (DNA) damage, autophagy [17,55,56]. In the following, we will briefly focus on the most common risk factors associated with NAFLD-related HCC with possible applicability for risk stratification in clinical practice.

### 3.1. Severity of Liver Disease

The primary risk determinant for HCC development in patients with non-cirrhotic NAFLD is the extent of liver fibrosis [12,57]. Risk stratification in HCC is significantly influenced by hepatic fibrosis [56], as cirrhosis is present in over 80% of patients with HCC [58]. In a recently published study, the likelihood of developing HCC surpassed the accepted thresholds for HCC surveillance (0.02/1000 person-years; hazard ratio 7.62; 95% confidence interval, 5.76–10.09) [49]. In a comparable manner, HCC incidence increased along with the progression of NAFLD, from simple steatosis (0.8 per 1000 person-years) to non-cirrhotic fibrosis (2.3 per 1000 person-years) and NAFLD-related cirrhosis (6.2 per 1000 person-years), respectively [12].

Patients with cirrhosis and probably advanced fibrosis (≥F3) continue to maintain an enduring susceptibility to HCC, and this aspect will be highlighted in the subsequent sections.

As outlined in the following pages, liver fibrosis can be assessed or predicted using histology, or non-invasive tests, such as liver stiffness measurement (LSM) [59].

### 3.2. Metabolic Syndrome and Metabolic Dysfunction Features

Metabolic dysfunction has assumed a prominent role in the pathogenesis of fatty liver disease [60,61]. In metabolic-associated fatty liver disease (MAFLD) patients, each metabolic feature was significantly associated with an elevated risk of MAFLD-HCC, while obesity presented the strongest correlation (adjusted odds ratio 3.63, 95% confidence interval 1.99–6.62) [62]. The metabolic syndrome (MS) components [63] nearly double the risk of HCC in the absence of overweight or obesity [64], and MS is considered to have an intricate linkage with NAFLD [65].

#### 3.2.1. Obesity

Obesity stands as the most common and well-described risk factor for NAFLD and is associated with a 1.5–4.5 times higher risk of HCC [66,67]. Among patients with NAFLD, diabetes (OR = 2.39, 95% CI: 2.04–2.79), metabolic syndrome (OR = 1.73, 95% CI: 1.49–2.01), and obesity (OR = 1.62, 95% CI: 1.43–1.85) displayed a significant association with an increased likelihood of HCC [68]. Although NAFLD is observed predominantly in persons with obesity, an estimated 7%–20% of individuals present lean-NAFLD [69,70], and this particularity should be considered when evaluating the likelihood of NAFLD-related HCC. Currently, standard weight-loss interventions and physical activity are strongly recommended to improve NAFLD-related outcomes [71], and embracing the Mediterranean diet could improve outcomes in NAFLD as well [72,73]. Nevertheless, future studies are needed to further examine the effect of lifestyle interventions in NAFLD- related HCC.

#### 3.2.2. Type 2 Diabetes Mellitus

Type two diabetes mellitus (T2DM) is a prevalent and firmly-established risk factor for NAFLD and is associated with an increased risk of liver disease progression [74,75,76]. Several studies have demonstrated that T2DM is associated with an elevated risk of NAFLD-related HCC [54]. Conversely, patients with diabetes have a two- to three-times higher risk of developing HCC, irrespective of the underlying liver disease and its severity [77,78,79]. As evidence, the relationship between diabetes and HCC exhibits bilateral characteristics: advanced liver disease promotes the onset of diabetes, and HCC stands as an important cause of death in this particular group of patients [80]. As such, optimal management of diabetes is essential in patients with NAFLD [29], and further actions should be taken to enhance HCC surveillance in diabetic patients.

#### 3.2.3. Hypertension, dyslipidemia

Hypertension and dyslipidemia might be independent risk factors for NAFLD-related HCC [79]. In a recently published study, including 1234 patients with putative metabolic conditions and NAFLD, patients with higher liver fibrosis scores who subsequently developed HCC had lower HDL-cholesterol (HDL-c) levels at time 0 (*p* < 0.001), and the authors concluded that HDL-c could become a novel marker to predict HCC in patients with NAFLD [81]. Hyperlipidemia might be associated with increased HCC incidence in noncirrhotic patients with biopsy-proven NASH [82]; nevertheless, these data should be further confirmed.

### 3.3. Older Age

With the increase in life expectancy, a higher number of individuals are experiencing age-related chronic metabolic disorders [83,84]. Although progression to cirrhosis seems to be similar among all ages in NAFLD, it has been recently demonstrated that the presence of liver-related events (LREs) increased along with advancing age [85]. In a retrospective study that comprised 31,505 patients with NAFLD, the 10-year risk of incident cirrhosis was similar between the pre-defined age groups (3.4% in age 18 to <40 vs. 3.7% in age 40 to <60 vs. 4.7% in age ≥ 60; *p* = 0.058), but the predictors of LREs, including the onset of HCC, were diabetes and age. Patients with ≥60 years exhibited the highest 10-year risk of LREs (0.2% in age 18 to <40 vs. 0.7% in age 40 to <60 vs. 1.1% in age ≥60; *p* = 0.008) [85].

### 3.4. Increased Liver Enzyme Levels

It has been recently demonstrated that patients with liver steatosis and elevated alanine transaminase (ALT) levels associate an increased risk of developing cirrhosis (adjusted hazard ratio 3.37; 95% confidence interval 2.34–4.86; *p* < 0.01). They also presented a higher tendency to develop HCC, even though it did not eventually reach statistical significance [86]. In an Asian population comprising 129,374 adults who participated in a health screening program, individuals with NAFLD and increased liver enzyme levels were associated with higher risk for both cirrhosis (adjusted hazard ratio 3.51, 95% confidence interval 2.36–5.22) and HCC (adjusted hazard ratio 1.91; 95% confidence interval 1.08–3.38) [87]. The authors concluded that NAFLD patients with increased liver enzyme levels should be monitored and guided in terms of behavioral modifications [87].

### 3.5. Alcohol and Tabaco Use

Alcohol and tobacco use increase the risk of many cancers, including HCC [88], and a recently published expert review stated that smoking and alcohol cessation should be considered important goals in the prevention of NAFLD-related HCC [89]. Currently, the patterns of alcohol consumption among patients with NAFLD and their implications in the wide context of the disease are controversial. A recently published study including 276 consecutive patients evaluated the impact of current and lifelong alcohol consumption in subjects fulfilling the criteria of NAFLD [90]. The authors illustrated that very low alcohol consumers (<70 g/week) showed a reduced occurrence of cirrhosis and HCC compared to abstainers and moderate consumers. We believe that alcohol drinking patterns should always be investigated in patients with suspicion of chronic liver diseases, and the potentially protective effect of very low alcohol consumption in the context of a Mediterranean diet in selected NAFLD patients should be further addressed.

### 3.6. Genetic Predisposition

In a recently published study, the presence of the PNPLA3 risk allele was increased in NAFLD patients exhibiting HCC [29]. Additionally, polygenic risk scores (PRS) might contribute to risk stratification in NAFLD-related HCC [91]. The high-risk variants PNPLA3-MBOAT7-TM6SF2-GCKR have been combined in a PRS, and its association with HCC was evaluated in patients with cirrhosis [92]. The PRS alone had modest discriminatory ability (c-statistic 0.58; 95% confidence interval 0.52–0.63); however, adding PRS to a predictive model with traditional HCC risk factors performed better (c-statistic 0.70; 95% confidence interval 0.64–0.76), increasing from 0.68 without the PRS (*p* = 0.0012). In a cohort of 2566 NAFLD patients evaluated for suspected liver disease or who underwent liver biopsy during bariatric surgery, of which 226 with HCC, variants in PNPLA3-TM6SF2-GCKR-MBOAT7 were combined in a hepatic fat PRS (PRSHFC), and then adjusted for HSD17B13 (PRS-5). A causal relationship between the presence of hepatic fat and HCC was supported by this evidence, and the authors concluded that the PRS improved the accuracy of HCC detection and it might further serve as an HCC risk stratification tool in patients with dysmetabolism. At present, the clinical utility of a genetic-based approach for NAFLD-related HCC risk stratification is not supported by strong evidence or cost-effectiveness [93].

### 3.7. Other Aspects Related to Risk Stratification in NAFLD-Related HCC

When screening for NAFLD-related HCC, other aspects, such as race and ethnicity [94], gender differences, and hormone/menopause status [95], should be taken into consideration. Together with clinical suspicion, these factors play an important role in fostering personalized medicine and risk assessment on a case-by-case basis. In a recently published study, among the first-degree relatives of Mexican Americans with HCC, 19 (17%) presented significant fibrosis and 47 (42%) definite hepatic steatosis, thus highlighting the importance of familial, cultural, and geographical aspects when screening for liver diseases such as NAFLD, and HCC. In the same study, diabetes (odds ratio 3.2, 95% confidence interval 1.1–9.2; *p* = 0.03) and elevated aspartate aminotransferase (AST) levels ≥30 u/L (odds ratio 4.0; 95% confidence interval 1.4–11.7; *p* = 0.01) predicted the presence of significant fibrosis [96]. Coffee consumption has been associated with decreased risk of developing HCC in patients with chronic liver diseases [36,43]; nevertheless, there are insufficient data to recommend a specific dose [97,98].

Figure 1 summarizes some of the risk factors associated with NAFLD-related HCC.

## 4. Non-Invasive Assessment of Liver Fibrosis for HCC Risk Stratification in NAFLD

The traditional approach to diagnosing NAFLD and NASH entails conducting a liver biopsy (LB) [99,100,101]. Nevertheless, LB is indicated with caution, as it holds the potential for adverse effects [102]. The limitations of conducting an LB, such as sampling errors, intra/inter-observer variability, and limited repeatability over time, have led to the development of non-invasive tests (NITs). NITs alone, or integrated into algorithms, could help the clinician to better evaluate and stratify patients at risk of developing liver-related events (LREs), including NAFLD-related HCC while pursuing cost-effective surveillance approaches [103,104,105]. Traditionally, these non-invasive methods, which have the potential of assessing different features of the probably damaged liver parenchyma, such as fibrosis or steatosis, rely on two different approaches: a ‘‘biological’’ approach based on serum biomarkers levels and a ‘‘physical’’ approach based on liver stiffness (LS) assessment [106]. Considering the impracticality of liver biopsies due to the high number of patients with NAFLD and their associated limitations, multiple non-invasive tests (NITs) have become widely used for staging fibrosis in NAFLD [107]. The higher the stage of liver fibrosis, the greater the likelihood of HCC development. Nevertheless, it remains important to further understand how to interpret the values and how to integrate them in multistep, cost-efficient, and accurate algorithms for HCC risk stratification purposes, especially in non-cirrhotic NAFLD.

### 4.1. Non-Invasive Tests (NITs) and Their Applicability in NAFLD-Related HCC Screening and Risk Stratification

Many serum biomarkers have been proposed for staging liver fibrosis in NAFLD. Currently, the Fibrosis-4 index (FIB-4) and NAFLD fibrosis score (NFS) are the most extensively utilized for this purpose [107].

#### 4.1.1. Performance of NITs for Liver Fibrosis Assessment in NAFLD

FIB-4 (Fibrosis-4 Index, comprising of PLT—platelets; AST—aspartate aminotransferase; ALT—alanine aminotransferase) was primarily validated as an accurate marker of liver fibrosis in patients with hepatitis C [108]. Over time, its usefulness was expanded to other liver disease etiologies as well as various clinical settings. A FIB-4 < 1.3 ruled out the presence of advanced fibrosis in patients with NAFLD with high sensitivity (Se) and negative predictive (NPV) values of ≥90%, proving its usefulness in stratifying patients for liver disease severity [109,110,111]. This inferior FIB-4 cut-off could be adapted to age for better accuracy, as a cut-off of <2.0 to exclude advanced fibrosis in patients over 65 years led to a decrease in grey-zone results [112]. The superior FIB-4 cut-off of ≥ 2.67 for ruling in advanced fibrosis was strongly associated with all-cause mortality and liver-related events among NAFLD patients [113,114], while values of ≥3.25 applied for the same purpose, reduced the number of patients in the grey zone (1.30–3.25) [115].

#### 4.1.2. Performance of NITs for NAFLD-Related HCC Risk Stratification

NITs could be relevant for enhancing risk stratification or predicting the occurrence of LREs, including HCC, in the setting of NAFLD [116]. The decision regarding NITs’ applicability in both primary care and more specialized services is governed by distinct features such as the particularities of the clinical setting, the pre-test probability of the disease/condition, availability, test performance, and cost. Simple fibrosis scores such as FIB-4 and NFS have the potential to be applied in routine clinical practice in an efficient manner and with optimal cost management [117].

Recently, in the setting of NAFLD-related HCC, the prognostic value of FIB-4 was evaluated in a very large European sample comprising 18 million individuals. In this patient population, the risk of developing HCC increased progressively with the augmentation of FIB-4. Those patients with a score between 1.30–2.67 had a risk ratio for HCC of 3.74, and those with a score >2.67 of 25.2, when compared to the patients with FIB-4 < 1.3 [118].

In a recently published study, the prognostic value of FIB-4 for HCC risk stratification in cirrhotic-NAFLD was evaluated among 122 consecutively included individuals [119]. Compared to the predefined FIB-4 inferior cut-off of 1.3, the 1.45 cut-off allowed the ruling out of a greater number of patients with minimal risk of HCC (23 vs. 18 patients). Regarding the superior FIB-4 cut-offs, the cumulative incidence of HCC using the 3.25 threshold (rule in) was distinctly higher than the 2.67 cut-offs (19.4% vs. 13.3%), and these cut-offs were independently associated with HCC development after adjusting for sex, BMI and T2DM (hazard ratio 6.40; 95% confidence interval 1.71–24.00; *p* = 0.006). The authors concluded that FIB-4 values of <1.3 and >3.25 could enable the risk stratification of long-term HCC development in cirrhotic individuals with NAFLD.

In another study that tested the prognostic performance of various NITs such as NFS, FIB-4, BARD, and APRI to predict HCC development in 1173 patients with NAFLD, the NFS significantly outperformed the other NITs (C-index: 0.901 ± 0.0302; AUROC = 0.889 ± 0.048) [120].

A recently published study, including 202,319 Veterans with NAFLD, demonstrated that the longitudinal changes in FIB-4 are associated with the risk of developing cirrhosis and HCC [121]. Among the 473 patients that were included in this study, the incidence rate of HCC was 0.28 per 1000 person-years (95% confidence interval 0.26–0.30). The authors highlighted that a high FIB-4 value of >2.67 at baseline and at 3 years of follow-up was linked to a >50-fold higher risk of developing HCC when compared to persistently low FIB-4 values (<1.45).

We believe that simple prediction NITs such as FIB-4 and NFS could be integrated in the near future in decision-making strategies to better stratify the NAFLD population that could benefit from HCC screening and could be included in surveillance programs. FIB-4, individually or integrated within multistep algorithms, may serve as a cost-effective and readily applicable test for healthcare professionals to identify NAFLD patients who are at low risk of progressing to cirrhosis and HCC.

### 4.2. Ultrasound Elastography and Its Applicability in NAFLD-Related HCC Screening and Risk Stratification

All hepatic disorders, whether focal or diffuse, are linked to modifications in the tissue’s structure, resulting in variations in the liver’s biomechanical characteristics. These variations can be quantified using different elastography techniques, such as (MRI)-based elastography techniques and ultrasound (US)-based elastography techniques. The quantitative ultrasound (US) elastography techniques with greater applicability in evaluating diffuse liver conditions are vibration-controlled transient elastography (VCTE), 2D shear wave elastography (2D-SWE), and point shear wave elastography (pSWE), VCTE being the most validated elastography technique in the field of liver diseases.

The prognostic role of liver stiffness measurement (LSM) using both US and MRI elastography stands as an area of active and vibrant research [122]. According to some experts, patients with NAFLD and NITs indicating advanced liver fibrosis or cirrhosis should be considered for HCC screening [29]. So far, both US (especially VCTE) and MRI elastography techniques have been evaluated in NAFLD patients for estimating the risk of HCC development [122]; nevertheless, the level of evidence is still limited, especially for non-cirrhotic NAFLD.

Vibration-controlled transient elastography (VCTE) was the first tool used to measure and investigate liver stiffness (LS), and it also stands as the most validated elastography technique, with the largest amount of data available, in NAFLD and NASH [123,124]. As the tissue becomes stiffer, LS values increase, whereas lower values suggest a more elastic liver. Current practice guidelines state that an LS of around 4.5–5.5 kPa is characteristic of a healthy population [125].

#### 4.2.1. Performance of Vibration Controlled Transient Elastography for Liver Fibrosis Assessment in NAFLD

Important results considering the performance of VCTE in patients with NAFLD for the assessment of liver fibrosis came from a recently published meta-analysis [126]. sAUC values for VCTE were as follows: for detecting any stage of fibrosis (≥F1)—0.82 (95% CI 0.78–0.85, sSe 78%, sSp 72%), for significant fibrosis (≥F2)—0.83 (95% CI 0.80–0.87, sSe 80%, sSp 73%), for advanced fibrosis (≥F3)—0.85 (95% CI 0.83–0.87, sSe 80%, sSp 77%), for cirrhosis—0.89 (95% CI 0.84–0.93, sSe 76%, sSp 88%). For diagnosing advanced fibrosis (≥F3), the point of maximum Youden index was achieved with a threshold of 8.7 kPa. A cut-off of 8.9 kPa showed a sSe of 80% and a sSp of 77%, while a cut-off of 9.5 kPa exhibited a sSe of 76% and a sSp of 80%. Nevertheless, the study did not include patients with morbid obesity, and some of the included studies did not use the XL probe.

The recently released Baveno VII guidelines [127] introduced significant enhancements to the applicability of LSM by VCTE in differentiating compensated advanced chronic liver disease (cACLD)—the spectrum of advanced (severe) fibrosis and cirrhosis. The guidelines indicate that LS values below 10 kPa in the absence of additional clinical/imaging features exclude cACLD, values of 10–15 kPa are suggestive of cACLD, and values above 15 kPa are highly suggestive of cACLD. Nevertheless, patients with LS values of 7–10 kPa and ongoing liver damage should be individually monitored for any changes suggesting the progression to cACLD [127].

The guidelines also introduced “the rule of five” (10–15–20–25 kPa), which can function as a gauge for escalating relative risks of decompensation and liver-related mortality in chronic liver disease, irrespective of the underlying etiology. For prognostic purposes, in a recently published study, a VCTE cut-off of ≥16.6 kPa predicted the progression from bridging fibrosis (F3) to cirrhosis, and a cut-off of ≥30.7 kPa predicted the development of LREs among cirrhotic patients [128].

By using VCTE for liver fibrosis assessment, NAFLD patients could be categorized into three risk groups: <8.0 kPa (indicating a low risk of ≥F3), 8.0–12.0 kPa (reflecting an intermediate risk of ≥F3), and >12.0 kPa (signifying a high risk of ≥F3), while a cut-off of ≥ 20 kPa could serve as a reliable diagnostic indicator of cirrhosis (Sp 95%) in the absence of a liver biopsy [129,130].

#### 4.2.2. Performance of Vibration Controlled Transient Elastography for NAFLD-Related HCC Risk Stratification

Among the US elastography techniques, the existing literature predominantly emphasizes VCTE as one of the most prominent elastography methods for LS assessment in NAFLD. Important aspects related to the performance of VCTE in predicting NAFLD-related HCC are synthesized in Table 1.

To note, there was a direct correlation between the risk of developing HCC and the baseline LSM by VCTE. In a population of NAFLD patients diagnosed by the US, HCC incidence increased from 0.32% for LS < 12 kPa, to 0.58% for LS between 12–18 kPa, 9.26% for LS between 18–38 kPa, and 13.3% for LSM > 38 kPa [138].

A recent study involving individuals with T2DM and NAFLD who underwent VCTE-LSM at baseline and were followed for an average duration of 50 months revealed significant findings on the relationship between baseline LS and the occurrence of LREs. Decompensation or primary liver cancer occurred in 6 out of 35 (17.1%) patients with baseline LS > 13 kPa, and the rate of HCC development in this particular subgroup was 2.1% per year. After adjustment for age and known cirrhosis before recruitment, individuals with LS > 13 kPa exhibited a 27.4-fold higher probability of experiencing a liver-related event (LRE) compared to those with LS < 13 kPa (95% CI, 7.86–95.50; *p* < 0.001) [134].

In a recently published study, the change in LSM by VCTE (difference between follow-up and baseline LSM, Δ-LSM) was independently associated with the following events: hepatic decompensation (hazard ratio 1.56; 95% confidence interval, 1.05–2.51; *p* = 0.04), HCC development (hazard ratio, 1.72; 95% confidence interval, 1.01–3.02; *p* = 0.04), overall mortality (hazard ratio, 1.73; 95% confidence interval, 1.11–2.69; *p* = 0.01), and liver-related mortality (hazard ratio, 1.96; 95% confidence interval, 1.10–3.38; *p* = 0.02). The authors concluded that a Δ-LSM >20% could be used to predict a high risk of HCC, liver decompensation, liver-related and overall death in patients with compensated advanced chronic liver disease (cACLD) [136].

In 1057 patients with NAFLD at baseline, FIB-4 and VCTE showed good accuracy for the prediction of LREs, including HCC, with Harrell’s C-indexes >0.80 (0.817 (0.768–0.866) vs. 0.878 (0.835–0.921), respectively, *p* = 0.059). VCTE presented a tendency to higher Harrell’s C-index than FIB-4, but without reaching statistical significance (0.878 (0.835–0.921) vs. 0.817 (0.768–0.866), *p* = 0.059) in this particular study. In the same light, the superior FIB-4 cut-off of >3.25 was associated with a 30-fold increased risk of LREs (adjusted hazard ratio 29.5; 95% confidence interval 10.6–82.3) and VCTE values above 12.0 kPa were associated with a 21-fold increased risk of LREs (adjusted hazard ratio 20.5; 95% confidence interval 4.9–86.5) [135].

Lee et al. developed a model for HCC prediction in NAFLD after testing its performance in both training and validation cohorts, as presented in Table 1. In this study, the diagnosis of NAFLD was established with the controlled attenuation parameter (CAP) by VCTE. During the follow-up, 22 patients (0.8%) in the training cohort developed HCC. The NAFLD-related HCC predictive model included the following variables: age ≥ 60 years (HR 9.1), aspartate aminotransferase (AST) ≥ 34 IU/L, platelet count <150 10^3^ µ/L (HR 3.7), and LS ≥ 9.3 kPa (HR 13.8). The AUROCs for HCC prediction were calculated at 2, 3, and 5 years of follow-up and were: 0.948, 0.947, 0.939 in the training cohort and 0.777, 0.781, and 0.784 in the validation cohort, respectively. The authors concluded that the new risk prediction model for NAFLD-related HCC showed acceptable performance [137].

Another cross-sectional study that included 191 NAFLD individuals with comorbidities investigated the role of different US-based NAFLD scoring systems in narrowing the group at high risk of developing HCC. For this purpose, the patients were stratified into three risk groups on the base of FIB-4. After the first selection obtained by FIB-4, Agile 3+ proved to be useful in further narrowing the number of patients at risk of developing HCC, with high Se and high NPV (26 HCC among 80 patients [33%]) [141].

It has been recently demonstrated that in biopsy-proven NAFLD patients (2518 included in an individual patient data meta-analysis with a median follow-up of 57 months), with the composite endpoint including the development of HCC, validated NITs such as LSM-VCTE, FIB-4, and NFS demonstrated comparable prognostic accuracy to histologically evaluated liver fibrosis (F0–2 vs. F3 vs. F4) [23], and also provided very similar prognostic information relied on literature-based cutoffs [142]. The cut-offs considered in the above-mentioned meta-analysis were the following: <10 kPa vs. 10 to <20 kPa vs. ≥20 kPa for LSM-VCTE [129]; <1.3 vs. 1.3 to 2.67 vs. >2.67 for FIB-4 [111]; <–1.455 vs. –1.455 to 0.676 vs. >0.676 for NFS [143]. Log-rank tests comparing event-free survival based on histology and LSM-VCTE suggested significant differences between patient subgroups stratified using the validated cut-offs (*p* < 0.0001 for histology; *p* < 0.0001 for LSM-VCTE). Considering the high number of patients that were included in this individual patient data meta-analysis, the authors concluded that compared to previous results [135], their study would suggest that FIB-4, LSM-VCTE, and histologically assessed liver fibrosis perform equally well.

#### 4.2.3. Performance of the Controlled Attenuation Parameter (CAP-VCTE) for NAFLD-Related HCC Risk Stratification

Regarding the prognostic value of the controlled attenuation parameter (CAP) measurement using VCTE, the existing literature is relatively limited and exhibits contradictory findings [144]. A CAP value of >220 dB/m was independently associated with an increased risk of developing relevant clinical events [145], while in another published study, neither the presence nor the severity of liver steatosis as measured by CAP predicted the development of HCC, the occurrence of either LREs or cardiovascular events [140]. These findings align with the most recent outcomes reported by Scheiner et al. [146]. Therefore, further research is necessary to clarify the prognostic role of CAP and the applicability of the latest CAP update (the continuous attenuation parameter) in NAFLD patients, including its potential association with the development of HCC, or other LREs.

According to some authors, patients with NAFLD and VCTE-LSM values above 15 kPa, should undergo HCC screening, as the annual incidence of HCC in this particular subgroup is greater than the cost-effectiveness threshold of 1.5% typically used for HCC screening in cirrhosis patients. [147].

A recent clinical practice update from the American Gastroenterological Association (AGA) regarding HCC in individuals with NAFLD suggested that HCC screening should be considered in patients where non-invasive markers suggest advanced fibrosis (≥F3) or cirrhosis. Consequently, the authors suggested the utilization of a combination of at least two non-invasive testing modalities, such as a serum-based NIT together with an elastography-based NIT (point-of-care testing), for improved HCC risk stratification [29].

We believe that well-identified patients with NAFLD, such as those with advanced fibrosis (≥F3), with or without cumulative risk factors for HCC development, could be screened for HCC in the future. Literature-based thresholds, such as the ones that are currently used for FIB-4 and VCTE or newly validated ones, could be applied to better stratify the risk of HCC development in NAFLD patients. To minimize the likelihood of misclassification, combining two or more NITs of separate categories (i.e., blood-based, imaging-based), and assessing for their agreement, could be of interest. NITs could facilitate the development of risk-stratification algorithms, clinical care pathways, and surveillance protocols for NAFLD-related HCC. An outlook on this aspect is presented in Figure 2.

## 5. Risk Stratification Biomarkers and Scores with Potential Applicability in NAFLD-Related HCC

According to the currently available practice guidelines, ultrasound (US) with or without serum alpha-fetoprotein (AFP) levels is the recommended tool for HCC surveillance. Besides US and AFP, a substantial number of clinical prediction scores are available to estimate the stage of fibrosis, grade of steatosis, or the risk of HCC in NAFLD patients, with different levels of clinical utility and validation [148,149]. At present, none of these calculators are prepared for implementation on guiding HCC surveillance; nevertheless, novel and efficacious risk stratification biomarkers may hold the potential to identify specific subgroups of non-cirrhotic NAFLD who would derive benefits from HCC surveillance [150]. At present, these tools could serve as supplementary resources in deciding whether to initiate or defer HCC screening. Some of them are briefly presented in Table 2.

## 6. Ultrasound for Hepatocellular Carcinoma Surveillance

At present, society guidelines recommend HCC primary surveillance for liver cirrhosis, or at-risk patients with chronic hepatitis B (CHB), based on abdominal ultrasound (US) evaluation with or without serum alpha-fetoprotein (AFP) levels [34,35,36,159].

HCC surveillance is associated with improved survival in patients with Child–Turcotte–Pugh A or B cirrhosis but seems to have no benefit in most patients with Child–Turcotte–Pugh C cirrhosis—outside of liver transplantation—given the high competing risk factors of liver-related mortality [43,160]. HCC surveillance should be performed at semiannual (approximately every 6 months) intervals [161]. The recommendation was initially formulated based on the tumor doubling time [162,163], but it was subsequently demonstrated that semiannual surveillance is associated with earlier tumor detection and improved survival compared with annual surveillance [164].

By expert opinion, HCC surveillance with abdominal US with or without serum AFP levels twice per year is recommended for patients with lean NAFLD and clinical indicators consistent with liver cirrhosis [159].

In a recently published meta-analysis, when comparing the sensitivity of US with or without AFP for the detection of HCC at any stage, the sensitivity of US alone was 78% (95% CI 67–86%) compared to 97% (95% CI 91–99%) for US plus AFP [37]. An AFP cutoff of 20 ng/mL provided a sensitivity of ~60% and a specificity of ~90%, although the optimal cutoff might be lower in those with nonviral etiologies of cirrhosis [165]. Longitudinal changes in AFP may also increase test performance characteristics versus AFP interpreted at a single threshold, so patients with rising AFP on two consecutive tests or doubling of AFP levels may also warrant diagnostic imaging, but this strategy still requires validation for how it can be best implemented [166]. Diagnostic evaluation with multiphasic contrast-enhanced CT or MRI in patients with AFP ≥20 ng/mL or rising AFP is currently advised [43]. Elevated AFP levels can be observed in other cancers, including intrahepatic cholangiocarcinoma, gastric cancer, and germ cell tumors [167,168].

Interventional programs significantly improve the rate of HCC surveillance as patients were six times more likely to undergo abdominal US surveillance if they participated in interventional surveillance programs versus standard of care (OR = 6.00; 95% CI: 3.35–10.77) [41,169].

### Limitations of Ultrasound-Based HCC Surveillance in NAFLD Patients

US is an operator-dependent modality, showing significant variability across centers. It is also influenced by body habitus, which can lead to limited or inconclusive results and decreased performance in detecting early-stage HCC. Poor exam quality has been reported in up to 20% of ultrasounds, mainly in patients with high BMI and NAFLD [37,40].

US appears to be more inaccurate in detecting HCC in patients with NAFLD-associated cirrhosis compared to other etiologies of liver cirrhosis [170]. This is attributed to the fact that steatosis leads to increased ultrasound (US) attenuation, which hampers, among others, the detection of deep liver nodules. Furthermore, there is a correlation between increasing body mass index (BMI) and a reduction in US accuracy, which follows a relatively dependent pattern [171].

Patients with limited US visualization may undergo surveillance with superior imaging techniques such as contrast-enhanced magnetic resonance imaging (MRI) or multiphase computer tomography (CT) [43], but given radiation exposure with a CT scan, MRI seems safer for repeated testing. There is modeling evidence that MRI could be cost-effective for HCC surveillance in patients at high risk of HCC development [172]. For non-invasive diagnostic purposes of focal liver lesions, the LI-RADS algorithm has been developed and presents the capacity to assist in communication and clinical decision-making [173].

## 7. Future Perspectives

Innovative and novel approaches are needed to reduce the late diagnosis of chronic liver diseases and prevent their complications [174]. NAFLD-related HCC surveillance in non-cirrhotic patients is currently debatable, emphasizing the need for further interventions that could better identify individuals at risk and promote the completion of HCC surveillance [149]. In our opinion, developing algorithms that could identify non-cirrhotic NAFLD patients who would benefit from HCC screening is of great interest. Combining clinical features, routine, and simple blood-based NITs, together with more complex markers that have the capacity to reflect the dynamic process of liver fibrogenesis, all together have the potential to function as valuable prognostic instruments in NAFLD-related HCC risk stratification.

Considering, in an integrated approach, the presence of risk factors for HCC, the FIB-4 score, and well as sequential or combined LSM by VCTE or by other techniques, such as pSWE, or 2D-SWE [175,176], could lead to developing an innovative yet cost-effective algorithm, capable of identifying non-cirrhotic NAFLD patients who may benefit from HCC surveillance [177].

If confirmed in future studies, the prognostic capability of multistep or multimarker algorithms could establish them as fundamental tools for risk stratification and risk management in individuals with NAFLD [178,179]. Nevertheless, we should not forget about personalized screening approaches to improve early detection and management of HCC. Updates and new techniques, such as the recently introduced multiparametric ultrasound (US), hold promise for improving the management of NAFLD patients [180]. The non-invasive assessment of liver inflammation, steatosis, and fibrosis by the multiparametric US, if accurate and reproducible, could have the potential to identify patients at risk of disease progression and introduce novel advancements in risk stratification. This, in turn, could lead to more precise management strategies.

Individuals with both NAFLD and notable non-hepatic comorbidities, such as cardiovascular concurrent illnesses, may experience less benefit from HCC surveillance, given the increased risk of non-liver-related mortality. However, these factors need to be further evaluated [181].

The identification of novel, reliable, non-invasive biomarkers derived from metabolomics, proteomics, transcriptomics, or extracellular vesicles such as exosomes, holds promise as supplementary resources for both diagnosis and prognosis in NAFLD patients and NAFLD-related HCC, respectively [182,183,184].

Machine learning and the broader field of artificial intelligence could represent a roadmap to the future development of novel biomarkers, could predict liver-related events, including the development of HCC, and overall improve the assessment of key outcome changes in NAFLD [185,186,187,188].

In the event that a precise risk stratification method could identify a subgroup of NAFLD patients with advanced fibrosis and a comparable risk of HCC to those with cirrhosis, these individuals might benefit from better disease management [150].

The recently introduced terminology for NAFLD will definitely bring new opportunities and future perspectives, including advancements in HCC risk stratification. As the principal limitations of NAFLD and NASH are the reliance on exclusionary confounder terms and the use of potentially stigmatizing language, a new nomenclature, metabolic dysfunction-associated steatotic liver disease (MASLD), has very recently been proposed [61], and the implications of this change are yet to be fully explored [189].

## 8. Conclusions

Risk stratification is crucial for managing NAFLD patients, and this review highlights important aspects linked to the occurrence of HCC in both cirrhotic and non-cirrhotic NAFLD. In summary, the rapidly increasing prevalence of NAFLD and the associated risk of HCC development, especially among patients with non-cirrhotic NAFLD, underscores the need for novel risk stratification tools that could provide early HCC diagnosis and enhanced surveillance. To increase HCC detection, screening tool accuracy, implementation, and compliance should be improved.

In essence, this review outlines opportunities for the development of NAFLD-related HCC risk stratification strategies, incorporating both personalized and more structured approaches, to enhance HCC screening and surveillance effectiveness. Currently, it is still unclear which subset of non-cirrhotic NAFLD patients could benefit from HCC screening, and a better characterization and stratification of this population is definitely required. Although there is an increased risk of HCC among non-cirrhotic NAFLD, the precise pre-test probability in this population still requires further establishment. Nevertheless, we are confident that this review could assist the reader in making informed decisions regarding the screening for NAFLD-related HCC in a more systematized manner.

For risk stratification purposes, non-invasive tools of prognostic relevance, such as ultrasound elastography, could bring important future contributions, leading to improved outcomes in NAFLD and NAFLD-related HCC.

## Figures and Tables

**Figure 1 cancers-15-04097-f001:**
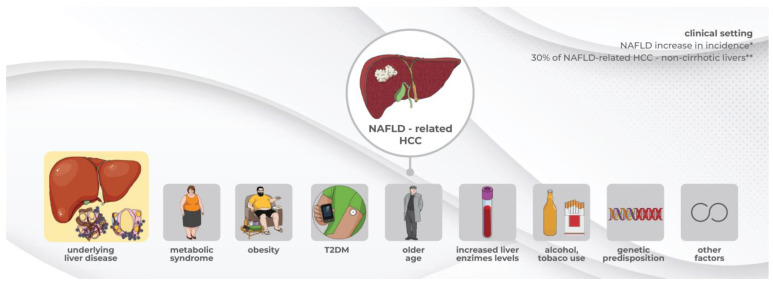
Factors increasing the risk of NAFLD-Related HCC. NAFLD—non-alcoholic fatty liver disease, HCC—hepatocellular carcinoma, T2DM—type 2 diabetes mellitus, * as published in the literature [1,2], ** as published in the literature [9,10,18].

**Figure 2 cancers-15-04097-f002:**
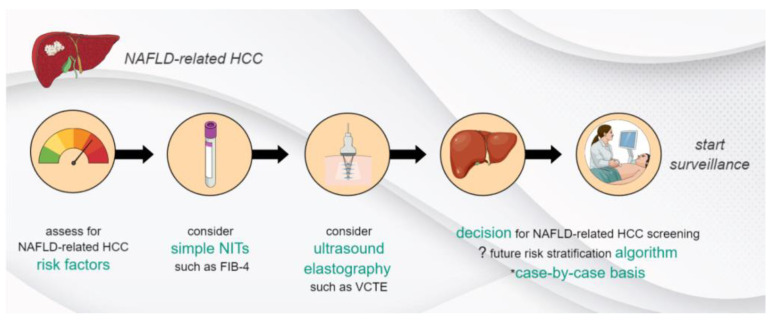
Outlook on developing risk stratification algorithms for NAFLD-related HCC. NAFLD—non-alcoholic fatty liver disease, HCC—hepatocellular carcinoma, NITs—non-invasive tests, FIB-4—Fibrosis-4 Index, VCTE—vibration-controlled transient elastography, ?—possible perspective, *—to be considered.

**Table 1 cancers-15-04097-t001:** Relevant aspects for VCTE in predicting NAFLD-related HCC.

Refs.	Study Population	LREs, Including HCC	Risk Stratification According to LSM	LSM and Risk of HCC, or LREs, Including HCC
Pennisi G, 2023 [131]	614 entire cohort 520 bp NAFLD 94 clinical NAFLD compensated cirrhosis	57 LREs 13 HCC	≥9.6 kPa-high risk of ≥F3; Agile 3+ > 0.68-high risk of ≥F3;	21.1% and 30.9% of pt defined as having high risk of ≥F3 by LSM and Agile 3+, respectively, developed LREs;
Davitkov P, 2023 [132]	13,629 NAFLD (ICD codes)	42 HCC	9.5–12.5 kPa-increased likelihood of ≥F3 with low likelihood of cirrhosis; 12.5–14.5 kPa-high likelihood of ≥F3 with some overlap of cirrhosis; >14.5 kPa-high likelihood of cirrhosis;	HCC incidence per 100 py (95% CI): < 9.5 kPa-0.04 (0.01, 0.1); 9.5–12.4 kPa-0.2 (0.05, 0.51); 12.5–14.4 kPa-0.58 (0.16, 1.49); ≥ 14.5 kPa-1.02 (0.68, 1.46);
Braude M, 2022 [133]	7079 NAFLD (ICD codes)	13 HCC primary 6 HCC comorbid COD in 8.8% of the DC	≥10 kPa-suggestive of cACLD;	LSM suggestive of cACLD associated with mortality (HR 2.31, CI 1.73–3.09, *p* < 0.001); increased LSM (HR 1.02 per kPa, CI 1.01–1.03, *p* < 0.001) associated with higher rates of all-cause mortality;
Johnson AL, 2022 [134]	243 NAFLD	3 HCC	8.0 kPa-clinically significant fibrosis; ≥9.5 kPa-to indicate ≥F3; >13 kPa-to indicate cirrhosis;	DLD or primary liver cancer occurred in 6 of 35 (17.1%) patients with baseline LSM > 13 kPa, during median fw 50 mo; for LSM > 13 kPa, rate of HCC or decompensation event is 2.1% per year;
Pons M, 2022 [57]	996 NAFLD 231 baseline cirrhosis	35 HCC	≤8.1-F0/F1 no or mild fibrosis; ≥8.2-≥F2 moderate fibrosis; ≥9.7-≥F3 severe fibrosis; ≥13.6-F4 cirrhosis;	HCC incidence: overall-9.49 (95% CI 6.4–13.9)/1000 py; non-cirrhotic-0.93 (95% CI 0.23–3.7)/1000 py; cirrhotic-41.2 (95% CI 27.6–61.6)/1000 py; cumulative incidence of HCC significantly higher in pt with cirrhosis, LSM (>8 kPa), and FIB-4 ≥1.3 in < 65 y and ≥2 in > 65 y);
Boursier J, 2022 [135]	1057 NAFLD	62 LREs 14 HCC	<8.0-low risk of ≥F3; 8.0–12.0-intermediate risk of ≥F3; >12.0-high risk of ≥F3;	similar risk of LREs for pts with “FIB4 <1.30”, respectively pts with “FIB4 ≥ 1.30 and VCTE <8.0 kPa” (adjusted HR 1.3; 95% CI 0.3–6.8); significantly increased risk of LREs for pts with “FIB4 ≥1.30 and VCTE 8.0–12.0 kPa” (aHR 3.8; 95% CI 1.3–10.9); higher risk of LREs for pts with “FIB4 ≥ 1.30 and VCTE >12.0 kPa” (aHR 12.4; 95% CI 5.1–30.2);
Petta S, 2021 [136]	1039 NAFLD with cACLD	35 HCC	improvement in LSM-reduction of more than 20%; stable LSM-reduction of 20% to an increase of 20%; impairment in LSM-increase of 20% or more;	LSM-independently associated with occurrence of HCC (HR, 1.03; 95% CI, 1.00–1.04; *p* = 0.003); Δ-LSM-significant predictor of HCC (HR: 1.72; 95% CI, 1.01–3.02; *p* = 0.04); improved LSM-low HCC risk; stable LSM-intermediate HCC risk; impaired LSM-high HCC risk;
Lee JS, 2021 [137]	NAFLD of any degree: 2666: training cohort 467: validation cohort	22 HCC training cohort	N/A	LS ≥ 9.3 kPa was independently associated with increased risk of HCC in a risk prediction model (HR = 13.757 (95% CI 2.826–66.955, *p* = 0.001), together with age ≥60 years (HR 9.1), AST > 34 IU/L, platelet count <150 × 103/µL (HR 3.7), LS ≥ 9.3 kPa (HR 13.8); AUC for HCC prediction: - training cohort, at 2, 3, and 5 years, respectively: 0.948, 0.947, 0.939; - validation cohort, at 2, 3, and 5 years, respectively: 0.777, 0.781, 0.784;
;Shili-Masmoudi S, 2020 [138]	2245 NAFLD of any degree	21 LREs, including HCC	>12.0 kPa-high risk of ≥F3;	HCC incidence increased with bl LSM: <12 kPa-0.32%; 12–18 kPa-0.58%; 18–38 kPa-9.26%; > 38 kPa-13.3%;
Izumi T, 2019 [139]	1054 CLD 258 NAFLD of any degree	88 HCC	N/A	for NAFLD subgroup, incidence of HCC development was significantly higher among LS ≥ 5.4 kPa + CAP ≤ 265 dB/m than among others (HR 8.91, 95% CI 1.47–67.97, *p* = 0.0192)
Liu K, 2017 [140]	4282 patients 1542 NAFLD of any degree	45 LREs 34 HCC	N/A	LSM independently predicted LREs (including HCC); CAP did not predict LREs; CAP ≥248 dB/m (as a categorical variable) on univariate analysis-protective for hepatic decompensations (HR 0.339, 95% CI 0.190–0.839, *p* = 0.015) and almost HCC (HR 0.485, 95% CI 0.240–0.980, *p* = 0.044); (as a continuous variable)-trend for CAP being protective for hepatic decompensations (HR 0.994, 95% CI 0.988–0.999, *p* = 0.017) and HCC (HR 0.995, 95% CI 0.990–1.000, *p* = 0.068)

Refs.—references, LREs—liver-related events, HCC—hepatocellular carcinoma, LSM—liver stiffness measurement, bp—biopsy-proven, py—person-years, CI—confidence interval, DC—deceased cohort, DLD—decompensated liver disease, fw—follow-up, mo—months, kPa—kilopascals, F0—no fibrosis, F1—mild fibrosis, F2—significant/moderate fibrosis, F3—advanced/severe fibrosis, F4—cirrhosis, pt—patients, y—years, HR—hazard radio, aHR—adjusted hazard ratio, cACLD—compensated advanced chronic liver disease, LSM—liver stiffness measurement, LS—liver stiffness, Δ-LSM was defined as the difference between follow-up and baseline LSM, N/A—not applicable, AUC—area under the ROC curve, AST—aspartate aminotransferase, FIB-4—Fibrosis-4 Index, bl—baseline, CAP—controlled attenuation parameter.

**Table 2 cancers-15-04097-t002:** Scores for early detection of hepatocellular carcinoma.

Score	Components	Utility
MRI-AST (MAST) Score [151]	AST, MRE, MRI-PDFF	noninvasively identifies at-risk NASH, accurately predicts major adverse liver outcomes, HCC, liver transplant, and liver-related death
GALAD Score [152]	gender, age, AFP-L3, AFP, DCP	high sensitivity for HCC detection in a cohort of patients with cirrhosis
Multitarget HCC blood test (mt-HBT) [50]	methylation biomarkers, AFP, gender	early stage HCC detection for patients undergoing HCC surveillance
Serum-protein-based prognostic liver secretome signature (PLSec) [153]	high-risk and low-risk associated serum proteins	stratification of patients with advanced liver fibrosis for long-term HCC risk
Cook Score [154]	VCTE, AST, ALT, platelets, INR	prediction of prognosis in patients with chronic liver disease and guidance of individualized surveillance strategy
Hepatocellular Carcinoma Early detection Screening (HES) algorithm [155]	AFP, rate of AFP change, age, ALP, platelets	detection of HCC in patients with cirrhosis of any etiology
HCC Risk Calculator for NAFLD-Cirrhosis [156]	age, gender, diabetes, BMI, platelets, serum albumin, AST/√ALT ratio	estimation of HCC risk in patients with NAFLD-cirrhosis
Kininogen-based algorithm [157]	Doylestown algorithm plus fucosylated kininogen	early HCC detection
Doylestown algorithm (DA) [158]	age, gender, ALT, ALK	HCC risk assessment, identification of early-stage HCC

MRE—magnetic resonance elastography, MRI-PDFF—magnetic resonance imaging derived proton density fat fraction, NASH—non-alcoholic steatohepatitis, AST—aspartate aminotransferase, ALT—alanine transaminase, AFP—alpha fetoprotein, AFP-L3—Lens culinaris agglutinin-reactive AFP, DCP—des-g-carboxy-prothrombin, HCC—hepatocellular carcinoma, VCTE—vibration controlled transient elastography, INR—international normalized ratio, BMI—body mass index, ALK—alkaline phosphatase.

## Data Availability

Not applicable.

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
