# Peer review of "Exploring Opportunities to Enhance the Screening and Surveillance of Hepatocellular Carcinoma in Non-Alcoholic Fatty Liver Disease (NAFLD) through Risk Stratification Algorithms Incorporating Ultrasound Elastography"

_cancers, 2023, doi:10.3390/cancers15164097_

Round 1

Reviewer 1 Report

In this review article, the authors explored and discussed the potential role of liver ultrasound elastography as a risk assessment tool for hepatocellular carcinoma (HCC) development in non-alcoholic fatty liver disease (NAFLD) and highlights the importance of effective screening tools for early, cost effective detection and improved management of NAFLD-related HCC.

Comments

This is an interesting review article. The reviewer has only some minor concerns as follows:

1. In Line 171 for section 3 - Risk Factors Associated with NAFLD-Related Hepatocellular Carcinoma, this part has already been reviewed and discussed in many literatures and review articles, such as Ueno et al., World J Gastroenterol. 2022 Jul 21;28(27):3410-3421 and Teng et al., Journal of Clinical and Translational Hepatology 2022;10(5):955-964, , and this is not the main focus of this review article, so the length of this section can be shortened.

2. In Table 1, it looks a bit complicated. The words of “author, year” in the leftmost column of the top row can be changed to “refs.”, and moves to the rightmost column in which only reference numbers are shown.

3. The description in the Keywords could be more concise, the full name and abbreviation can be presented alternatively.

Author Response

Response to reviewer 1

Exploring Opportunities to Enhance the Screening and Sur-veillance of Hepatocellular Carcinoma in Non-Alcoholic Fatty Liver Disease (NAFLD) Through Risk Stratification Algorithms Incorporating Ultrasound Elastography

The authors of the present manuscript wish to express their gratitude towards the Reviewers and Editorial Office that contributed significantly to the improvement of the article through their suggestions and expertise. Therefore, every comment within the revision was addressed (and also highlighted in the revised version) and the article was modified according to the Reviewer’s instructions.

In this review article, the authors explored and discussed the potential role of liver ultrasound elastography as a risk assessment tool for hepatocellular carcinoma (HCC) development in non-alcoholic fatty liver disease (NAFLD) and highlights the importance of effective screening tools for early, cost effective detection and improved management of NAFLD-related HCC. This is an interesting review article. The reviewer has only some minor concerns as follows:

Comment 1:  In Line 171 for section 3 - Risk Factors Associated with NAFLD-Related Hepatocellular Carcinoma, this part has already been reviewed and discussed in many literatures and review articles, such as Ueno et al., World J Gastroenterol. 2022 Jul 21;28(27):3410-3421 and Teng et al., Journal of Clinical and Translational Hepatology 2022;10(5):955-964, , and this is not the main focus of this review article, so the length of this section can be shortened.

Response 1: We thang Reviewer 1 for this important comment. We referred in our review to the previously published articles that discussed the relation between NAFLD and HCC and we re-structured Section 3 for a better understanding of the focus of our review and the length of this section was shortened.

Comment 2: In Table 1, it looks a bit complicated. The words “author, year” in the leftmost column of the top row can be changed to “refs.”, and moves to the rightmost column in which only reference numbers are shown.

Response 2: We changed the words “author, year” to “refs” in Table 1. However, to be easier for the reader to follow the citation, we decided to keep the name of the first author for the articles that are cited in Table 1.

Comment 3: The description in the Keywords could be more concise, the full name and abbreviation can be presented alternatively.

Response 3: We thank Reviewer 1 for this suggestion. We reformulated the keywords.

We wish to express our gratitude towards Reviewers  for their extremely helpful comments that significantly improved our manuscript.

Reviewer 2 Report

The paper explores a field which is very hot at the moment: how to optimize the stratification of NAFLD patients according to their risk to develop HCC with the purpose to define and organize the best cost-effective screening and surveillance strategies. 

Despite this very interesting subject, I have some concerns about the work:

- the first part of paragraph 1 "Introduction" (until line 79) is confused and needs extensive rewriting. I would add some data on the difference of NAFLD patients developing HCC according to the underlying presence of cirrhosis or not (ex HCC patients without cirrhosis are older, more frequently diabetic and so on). I would eliminate the consideration on MASLD as it is no more discussed in the paper, maybe authors could make just a brief consideration in the discussion or conclusion section. 

I do strongly agree with authors that "we need to better understand which patients with non-cirrhotic NAFLD have sufficient risk to warrant HCC surveillance and identify optimal screening tools that could guide the risk stratification and frequency of monitoring" (lines 92-94) as up to 30% of HCC cases develop in non-cirrhotic NAFLD livers, but the paper does not explore this aspect deeply enough.

In paragraph 3, risk factors associated with NAFLD-related HCC are listed: by definition, obesity and T2DM are components of metabolic syndrome, together with arterial hypertension and dyslipidemia, so I would discuss these factors in sub-paragraphs of metabolic syndrome. Again, I would not include in this section the concepts of MAFLD and MASLD as they were formulated to include a wider patient population in which the metabolic dysfunction may not be the only cause of liver disease. 

Section 4 contains an extensive description of biological and physical non-invasive markers of liver fibrosis (with a particular focus on US tecniques) and their efficacy in prediction NAFLD-related HCC. Actually all the data can be summarised in: the higher the stage of fibrosis, the higher the risk to develop HCC or LRE (which is pretty obvious).

I would aspect something more on that 30% of cases occurring in non-cirrhotics: how are we expected to be able to identify them? Something is discussed  in section 5 and table 2 but needs to be expanded.

In general, quality of English language is good. As I said above, the first part of the introduction is not clear and needs revision.

Author Response

Response to Reviewer 2

Exploring Opportunities to Enhance the Screening and Sur-veillance of Hepatocellular Carcinoma in Non-Alcoholic Fatty Liver Disease (NAFLD) Through Risk Stratification Algorithms Incorporating Ultrasound Elastography

The authors of the present manuscript wish to express their gratitude towards the Reviewers and Editorial Office that contributed significantly to the improvement of the article through their suggestions and expertise. Therefore, every comment within the revision was addressed (and also highlighted in the revised version) and the article was modified according to the Reviewer’s instructions.

The paper explores a field which is very hot at the moment: how to optimize the stratification of NAFLD patients according to their risk to develop HCC with the purpose to define and organize the best cost-effective screening and surveillance strategies.

Comment 1: The first part of paragraph 1 "Introduction" (until line 79) is confused and needs extensive rewriting. I would add some data on the difference of NAFLD patients developing HCC according to the underlying presence of cirrhosis or not (ex HCC patients without cirrhosis are older, more frequently diabetic and so on). I would eliminate the consideration on MASLD as it is no more discussed in the paper, maybe authors could make just a brief consideration in the discussion or conclusion section.

Response 1: We thank Reviewer 2 for this very important comment. We rewrote section 1 “Introduction”. We added the information regarding the newly proposed nomenclature (MASLD) in section 7 – Future perspectives.

Comment 2: I do strongly agree with authors that "we need to better understand which patients with non-cirrhotic NAFLD have sufficient risk to warrant HCC surveillance and identify optimal screening tools that could guide the risk stratification and frequency of monitoring" (lines 92-94) as up to 30% of HCC cases develop in non-cirrhotic NAFLD livers, but the paper does not explore this aspect deeply enough.

Response 2: We thank Reviewer 2 for this comment. We rephrased that part of the review.

Comment 3: In paragraph 3, risk factors associated with NAFLD-related HCC are listed: by definition, obesity and T2DM are components of metabolic syndrome, together with arterial hypertension and dyslipidemia, so I would discuss these factors in sub-paragraphs of metabolic syndrome. Again, I would not include in this section the concepts of MAFLD and MASLD as they were formulated to include a wider patient population in which the metabolic dysfunction may not be the only cause of liver disease.

Response 3: We thank Reviewer 2 for this very important comment. We restructured the section. We also updated Figure 1.

Comment 4: Section 4 contains an extensive description of biological and physical non-invasive markers of liver fibrosis (with a particular focus on US techniques) and their efficacy in prediction NAFLD-related HCC. Actually all the data can be summarized in: the higher the stage of fibrosis, the higher the risk to develop HCC or LRE (which is pretty obvious).

Response 4: We thank Reviewer 2 for this comment. We concluded better section 4 and explained why the emphasis on the non-invasive tests is important.

Comment 5: I would aspect something more on that 30% of cases occurring in non-cirrhotics: how are we expected to be able to identify them? Something is discussed  in section 5 and table 2 but needs to be expanded.

Response 5: We thank Reviewer 2 for this comment. We better highlighted this aspect in the Introduction.

 We wish to express our gratitude towards Reviewers  for their extremely helpful comments that significantly improved our manuscript.